# Management of Extra-Pelvic Varicose Veins of Pelvic Origin in Female Patients

**DOI:** 10.3390/jcm14082707

**Published:** 2025-04-15

**Authors:** Aleksandra Jaworucka-Kaczorowska, Roshanak Roustazadeh, Marian Simka, Houman Jalaie

**Affiliations:** 1Center of Phlebology and Aesthetic Medicine, 66-400 Gorzów Wielkopolski, Poland; 2Department of Vascular and Endovascular Surgery, University Hospital RWTH Aachen, 52074 Aachen, Germany; rroustazadeh@ukaachen.de (R.R.); hjalaie@ukaachen.de (H.J.); 3Department of Anatomy, Institute of Medical Sciences, University of Opole, 45-040 Opole, Poland; msimka@uni.opole.pl

**Keywords:** pelvic vein incompetence, pelvic venous disorders, extra-pelvic varicose veins of pelvic origin, pelvic escape points

## Abstract

Extra-pelvic varicose veins (VVs), originating from incompetent pelvic veins, present a significant clinical challenge, due to their complex anatomy, etiology, and symptomatology. This review aims at providing a comprehensive overview of the diagnostic and therapeutic strategies for these cases and emphasizes the importance of a tailored, evidence-based approach to the effective management of these varicosities, particularly regarding the interplay between the pelvic and extra-pelvic venous systems. Diagnostic workup should be multifaceted, incorporating patient-reported symptoms, physical examinations, and duplex ultrasound imaging. Specific diagnostic assessments include evaluation of the pelvic escape points and the transvaginal and transabdominal ultrasonography, to analyze venous hemodynamics and identify anatomical abnormalities in the pelvic floor and pelvis. In patients presenting with additional pelvic venous insufficiency (PVI)-related pelvic symptoms, advanced diagnostic techniques, such as cross-sectional imaging, venography, and intravascular ultrasound can be valuable to confirm and establish the appropriate treatment strategy. Since most patients with extra-pelvic VVs of pelvic origin do not report pelvic symptoms, minimally invasive procedures, using the “bottom-up” approach, such as ultrasound-guided foam sclerotherapy of the pelvic escape points and extra-pelvic VVs, or surgical ligation and miniphlebectomy for these incompetent veins, are usually sufficient. There are several advantages of these local procedures: they are simple, radiation exposure and injection contrast agents are avoided, they are convenient for the patient since they are performed on an outpatient basis, and they can be easily repeated, if required. When the “bottom-up” treatment fails and the extra-pelvic VVs recur quickly or the patient develops pelvic symptoms, management of the pelvic veins including embolization of the ovarian veins or stenting of the iliac veins should be considered. Careful patient selection is essential to avoid overtreatment and achieve optimal clinical outcomes.

## 1. Introduction

This review paper is primarily aimed at summarizing diagnostics and treatment for extra-pelvic varicose veins (VVs) of pelvic origin. Such varicosities, located in the vulva, perineum, and/or lower limb, are increasingly being diagnosed, as the knowledge on anatomy and pathophysiology of this clinical entity is improving. These varicosities are one of the clinical manifestations of pelvic venous disorders (PeVDs), resulting from pelvic vein incompetence (PVI) and pelvic venous hypertension. PVI can develop from the incompetence of the left or right ovarian vein, the internal iliac veins (IIVs), their tributaries, or combined insufficiency of pelvic veins. Although PVI is typically a primary pathology, it can also develop secondarily to non-thrombotic iliac vein lesions (NIVLs), such as May–Thurner syndrome (MTS) (resulting from compression of the left common iliac vein (CIV) by the right common iliac artery) or nutcracker syndrome (associated with compression of the left renal vein between the aorta and the superior mesenteric artery) [1,2,3]. The role of MTS and IIV reflux is increasingly recognized in the pathophysiology of extra-pelvic VVs originating from pelvic conditions [4]. PVI can also evolve because of post-thrombotic stenoses of the inferior vena cava (IVC) and/or CIVs, resulting in an overload of venous circulation in the territory of the IIV [1,2,3]. In addition to post-thrombotic stenoses of the iliac veins, these veins can also be compressed by endometriotic lesions, neoplastic tumors, or peritoneal adhesions. Such non-thrombotic etiologies should be considered during differential diagnosis for PVI [4].

There are several predisposing factors of PVI. Pregnancy is one of the most important ones. PVI is uncommon during the first pregnancy, but typically develops around the fifth month of the second pregnancy, with the risk increasing during following pregnancies. Hormonal changes during pregnancy, such as elevated levels of estrogen, progesterone, and relaxin, are believed to weaken venous walls, with subsequent dilatation of the pelvic veins and their incompetence. A higher blood volume and vascular capacity in the ovarian veins, which are increased during pregnancy up to 60-fold and can persist even postpartum, are thought to be contributing factors. Compression of the pelvic veins by a pregnant uterus can further contribute to this pathomechanism [5]. Additionally, anatomical variations in venous valve distribution may influence the development of PVI. Distally located veins typically have a higher number of valves. Notably, valves are present in only 10% of the IIVs. In the ovarian veins, valves are primarily located in the caudal third; however, 15% of the left ovarian veins and 6% of the right ones lack valves entirely [6]. Although a majority of PVI patients are asymptomatic, some can present with clinical symptoms. Chronic pelvic pain (CPP) is the most common symptom of PeVD. Such a pain is defined as a pain in the pelvis which lasts for at least 6 months, is severe enough to cause functional disability of the patient, and provokes her to look for medical treatment. To be considered chronic, such a pain does not necessarily occur every day, yet it can follow a regular pattern, for example being active during menstruation (dysmenorrhea), during sexual intercourse (dyspareunia), or appear as a prolonged postcoital pain [7]. According to a systematic review by the World Health Organization, the prevalence of CPP ranged from 4% to 43%, and PVI accounted for 16% to 31% of these cases. In addition to CPP, as many as 54% of patients with PeVD report menstrual disorders, such as menorrhagia and menometrorrhagia, and intermenstrual bleeding in up to 25% of cases. Such bleeding disorders are probably related to venous congestion in the pelvis [8].

PVI-associated venous hypertension results in an overload of pelvic veins, which in turn results in dilatations of these veins and collateral outflows. Consequently, varicosities of pelvic venous plexuses develop, comprising dilatations of the ovarian, uterine, vesical, or prostatic plexuses. Since the pelvis veins are connected to those of the perineum and the lower limb through the so-called “pelvic escape points”, located in the pelvic floor, venous refluxes from the pelvis can be transmitted to the veins of the vulva, perineum, and the lower extremities, causing extra-pelvic VVs of pelvic origin [1,2]. This can result in the appearance of vulvar and perineal VVs (Figure 1A), atypical lower extremity VVs (Figure 1B), or an incompetence of the saphenous veins, still with refluxes originating in the pelvis instead of an incompetent terminal valve of the saphenous vein [2]. Patients usually present with moderately enlarged VVs (C2, based on CEAP classification), not causing skin damage but accompanied by diffuse limb pain/heaviness, typically related to hormonal phases. There is a high probability of hormones passing into the leg circulation through the leaking points, as suggested by Asciutto, G. et al. [9]. They typically do not present with pelvic symptoms since venous hypertension is transmitted from pelvis to the extra-pelvic veins. Therefore, CPP is observed in less than 10% of patients with extra-pelvic VVs of pelvic origin [10]. In the study by Gibson, only 7% of such patients reported such symptoms as CPP or heaviness [11]. Pelvic symptoms, including CPP, are of particular importance, since according to the European Society for Vascular Surgery (ESVS) guidelines on the management of chronic venous disease, their management strategy depends on the presence of such symptoms [2].

Of note, extra-pelvic VVs are quite often of pelvic origin, yet their actual prevalence is not known, because many of these VVs remain undiagnosed due to their unusual location. Since the diagnostic assessment involves examination of the external genitals, patients are not eager to be thoroughly assessed unless they suffer from severe symptoms. Furthermore, most doctors are not willing to perform such an extended diagnostic workup. Nonetheless, the prevalence of lower limb VVs of pelvic origin among females can be as high as 16.5%, and 25.6% among those with recurrent VVs [12]. According to the literature, vulvar VVs are found in approximately 22% of pregnant women, in 10% of non-pregnant women, and up to 34% of women presenting with enlarged and incompetent venous plexuses in the pelvis [13,14].

Although extra-pelvic VVs of pelvic origin are not typically accompanied by pelvic symptoms, very often these patients complain of symptoms in the area affected by these VVs. Such complaints comprise local pain, discomfort, tenderness, heaviness, itching, bleeding, and thrombosis [1]. Additionally, vulvar VVs can be associated with dyspareunia and vulvodynia.

## 2. Diagnostics

Diagnostic workups for extra-pelvic VVs of possible pelvic origin, in addition to the standard ultrasonographic assessment of the lower extremity veins (of note, this examination should be performed in the upright body position), should primarily focus on the identification of the pelvic escape points. Such an identification is of paramount importance for the proper treatment of these varicosities [2,15]. A detailed anatomical description of these pelvic escape points can be found in our other paper [16].

It should be emphasized that in addition to VVs located in the external genital region, many lower limb VVs are related to pathological refluxes originating in the pelvic veins. Therefore, in addition to the assessment of pelvic escape points, standard ultrasonographic examination of the lower limb veins with correct evaluation of terminal valve of the saphenofemoral junction should be performed [17]. In a case of an incompetent great saphenous vein or anterior accessory saphenous vein with competent terminal valve, the search for incompetent pelvic escape points is mandatory. Sometimes such examination in not easy to perform; therefore, proper time should be dedicated to detailed ultrasonographic examination.

### 2.1. Duplex Ultrasound of the Pelvic Escape Points

Duplex ultrasonographic assessment of the pelvic escape points should be performed in the anti-Trendelenburg or upright body position, with the use of a linear probe. There are seven to eight pelvic escape points that are described in the literature. These pelvic escape points provide anatomical connections of the pelvic veins with those located in the vulva, perineum, gluteal area, and the lower extremity [16,18,19,20,21,22].

The most common escape points are the pudendal ones, which are found in 60–70% of females with VVs of pelvic origin [16,19,20,21,22]. Pudendal pelvic escape points are related to the internal pudendal vein, which runs in the pudendal canal (Alcock’s canal) alongside the inferior ramus of the ischium. Venous reflux coming from the pelvis, transmitted through the internal pudendal veins, is then directed toward their tributaries, the deep dorsal veins of the clitoris, the veins of the bulbs of vestibule, and the perineal veins. Following this anatomical pattern, there are three pudendal pelvic escape points that should be checked: the clitoral escape point on either side of the clitoris, the intermediate labial escape point located in the middle part of the labium majus, and the perineal escape point located in the posterior third of the labium majus (Figure 2) [16,19,20,21,22]. An ultrasonographic search for incompetent pudendal escape points should comprise all three above-described locations. Venous reflux that originates in the internal pudendal veins can also be transmitted to the inferior rectal veins, although such an abnormality is rarely found in patients presenting with PVI. It should also be remembered that since the veins of both labia majora are interconnected, the pudendal escape point can contribute to the development of both ipsilateral and contralateral VVs in the genital region and/or lower limbs. Therefore, both sides of the vulva, as well as both lower extremities, should be carefully evaluated by means of ultrasonography.

The inguinal escape point is the second most common escape point, occurring in approximately 20–35% of patients with extra-pelvic VVs originating in the pelvis [16,20,21,22]. Reflux from the pelvic veins can be transmitted through the veins of the round ligament of the uterus, which runs alongside the inguinal canal (Figure 3), potentially causing VVs in the mons pubis, vulva, and/or the lower limb. In order to reveal pathological reflux associated with this escape point, ultrasonographic examination should include inspection of the inguinal canal (area above the inguinal ligament and behind the aponeurotic part of the external oblique muscle). The anti-Trendelenburg position and Valsalva maneuver can be useful for provoking refluxes in this area.

The gluteal escape points are associated with the incompetent superior or inferior gluteal veins, which are tributaries of the IIVs. They are relatively rare anatomical locations of pelvic refluxes. It is estimated that the inferior gluteal escape point, which is located below the piriformis muscle, is the cause of about 4% of VVs of pelvic origin, while the superior gluteal escape point, located above this muscle, is associated with about 2% of such VVs. Reflux transmitted through these pelvic escape points is typically associated with VVs at the buttocks and/or posterior aspect of the thigh.

In addition, this outflow route is associated with the fetal pattern of the venous outflow from the lower extremity, with the axial vein being the main vein of the extremity. In most adults, the axial vein is replaced in this function by the femoral vein. Yet, in some adult individuals this anatomy is still present, and usually is asymptomatic. Nonetheless, in addition to ultrasonographic examination of the above-described escape point located in the gluteal region, and epifascial veins of the posterior part of the thigh, deep veins of the posterior fascial compartment of the thigh should also be screened. This particularly concerns veins in the proximity to the sciatic nerve. Normally, there are only tiny veins running alongside the sciatic nerve (the veins of the sciatic nerve), and no reflux can be detected in these veins. Yet sometimes there is a bigger vein (or veins) in this location. In such a case it is called the persistent sciatic vein. This vein can be asymptomatic but can also be associated with symptoms related to sciatic nerve irritation, such as pain, numbness, or a tingling sensation in the leg. An enlarged vein accompanying the sciatic vein can either drain into the profunda femoris vein or its tributaries or can connect to the veins in the gluteal region, including the above-described gluteal escape points, hence the need for ultrasonographic inspection of the veins in the proximity of the sciatic nerve, particularly in patients presenting with symptoms typical for the irritation of this nerve [18].

The obturator escape point, accounting for about 3% of extra-pelvic VVs of pelvic origin, is related to incompetent obturator veins, which are tributaries of the IIV [16,20,21,22]. These veins connect pelvic veins with those of the lower limb through the opening in the upper part of the obturator foramen. Reflux from pelvic veins can be transmitted through the obturator veins to the deep veins of the medial fascial compartment of the thigh, and to the superficial veins located at the medial aspect of the upper thigh. Ultrasonographic screening for refluxes associated with the obturator escape point should primarily focus on these locations.

### 2.2. Duplex Ultrasonography of Pelvic Veins

For the time being, it remains controversial if detailed diagnostics of the pelvic veins in patients who present with extra-pelvic VVs of pelvic origin, yet do not complain of pelvic symptoms (presented in the Section 1) is necessary. According to the ESVS guidelines on the management of chronic venous disease, an additional transabdominal and/or transvaginal ultrasonographic evaluation should be considered in patients with suspected extra-pelvic VVs with pelvic origin [2], whereas the U.S. guidelines do not advocate for such an extensive diagnostic workup if the patient does not report pelvic symptoms [23]. In patients presenting with pelvic symptoms, pelvic veins should be assessed, even if such symptoms can be caused by many non-vascular pathologies.

#### 2.2.1. Transvaginal Ultrasonography

Transvaginal ultrasonography (TVUS) is a noninvasive and highly effective diagnostic tool for evaluating pelvic veins, particularly in the context of CPP. It plays a critical role in excluding alternative causes of CPP, especially gynecological conditions, such as endometriosis, adenomyosis, ovarian cysts, and myomas. TVUS enables a more detailed assessment of venous pelvic abnormalities, such as pelvic congestion syndrome. A transvaginal approach significantly reduces the distance between the ultrasound probe and the pelvic structures [2,24]. The periuterine, periovarian, and perivaginal venous plexuses are much more visible (Figure 4). It also concerns distal parts of the ovarian veins and tributaries of the IIVs [25]. Furthermore, TVUS allows for hemodynamic assessment of the pelvic vein reflux evoked by the Valsalva maneuver or by distal manual compression in the ipsilateral iliac fossa (Figure 4B) [26].

For proper evaluation of the pelvic veins, TVUS should be performed in the anti-Trendelenburg or upright body position [3]. Based on the systematic review, the presence of enlarged pelvic venous plexuses and a vein wider than 5 mm crossing the body of the uterus, and joining the enlarged periuterine venous plexuses on both sides of the uterus, (Figure 4) are considered as diagnostic criteria of high sensitivity and specificity for PeVD [27]. It should be noted that the patients with PVI often present with other comorbidities that can also contribute to CPP. In comparison with the general population, due to pelvic congestion and estrogen overstimulation, PVI patients are more often diagnosed with ovarian cysts (Figure 4) [24].

#### 2.2.2. Transabdominal Ultrasonography

Transabdominal ultrasonography (TAUS), alongside TVUS, serves as the initial step in PeVD diagnosis. It enables a direct visualization and evaluation of the inferior vena cava, iliac veins, renal, and ovarian veins, as well as of the pelvic venous plexuses, including their hemodynamic characteristics [2]. The examination is typically performed after an overnight fast, in order to lower the number of intestinal gases, which can overshadow vascular structures and lower the accuracy of ultrasonographic examination. During the procedure, the patient is either positioned in the upright body position, or in the supine one with their head elevated at the 30° angle [15,27].

PVI is often associated with ovarian vein incompetence. According to a recent systematic review, an enlarged ovarian vein which is wider than 6 mm (Figure 5A) and a reversed caudal flow in the ovarian vein (Figure 5B) are the most indicative markers of PeVD (Figure 5) [26]. Intermittent reflux in the ovarian vein triggered by a manual compression in the ipsilateral iliac fossa or by the Valsalva maneuver indicates an isolated ovarian vein incompetence. By contrast, a spontaneous reflux, with continuous flow in the left ovarian vein suggests that the PVI etiology is secondary to an extravenous cause (usually, compression) or an intravenous one (usually, post-thrombotic) [2].

In addition, since this diagnostic modality allows for real-time hemodynamic assessment, TAUS offers a valuable and non-invasive tool for identifying non-thrombotic iliac vein lesions and internal iliac vein reflux (Figure 6). This examination assesses the inferior vena cava, iliac veins (common, external, and internal), common femoral, femoral, and deep femoral veins, left renal vein, and gonadal veins. IIV reflux assessment via Doppler ultrasound identifies abnormal venous flow, with pathological reflux often characterized by a venous diameter exceeding 8 mm and reflux duration greater than two seconds during postural or compression maneuvers [28,29].

The direct visualization of blood flow dynamics facilitates accurate differentiation between true lesions and anatomical variations, thereby minimizing false positive diagnoses. Furthermore, incorporating postural changes from the supine to upright position during the examination enhances diagnostic accuracy by revealing dynamic blood flow patterns and venous compression variations not apparent in the supine position. This positional shift leverages gravitational forces to accentuate or exacerbate venous compression, aiding in the diagnosis of conditions such as iliac vein compression syndrome and further reducing the likelihood of misdiagnosis [28,30].

### 2.3. Advanced Diagnostics

In patients with extra-pelvic VVs of pelvic origin and coexisting pelvic symptoms, which are potentially attributable to PVI, the initial diagnosis of PVI that is typically made through TVUS and/or TAUS should be further confirmed using advanced diagnostic methods. Cross-sectional imaging is the most commonly employed approach for this purpose. Additionally, assessment of the supra-inguinal venous outflow with the use of intravascular ultrasonography (IVUS) and catheter venography can offer a valuable tool for establishing the appropriate treatment strategy [2].

#### 2.3.1. Cross-Sectional Imaging

Computed tomography (CT) venography and magnetic resonance venography (MRV) are crucial for identifying dilatated ovarian veins, dilatated pelvic plexuses (Figure 7), and extrinsic compression. These diagnostic modalities can reveal coexisting thrombotic occlusions, collateral outflow networks, and also other potential non-venous pathologies contributing to the symptoms. However, these diagnostic methods have their own limitations. Accuracy of the assessment depends on the patient’s hydration status, while CT scanning is associated with a high dose of radiation. Additionally, MRV of pelvic veins is not easy to perform properly and requires the expertise to do it correctly and obtain images of diagnostic value [29]. Of note, compression lesions diagnosed by CT venography or MRV should be interpreted with caution, since these diagnostic modalities are used when the patient is in the supine body position. In this body position, a revealed compression is often positional and should be confirmed by other methods. Otherwise, there is a risk of overdiagnosing the patient.

#### 2.3.2. Catheter Venography and Intravascular Ultrasound

Catheter venography and IVUS are the main invasive imaging modalities, which are recommended for patients who are evaluated for thrombotic and non-thrombotic iliac vein lesions when a potential endovascular treatment is considered during the same procedure (Figure 8 and Figure 9) [2].

Dynamic IVUS assessment, incorporating such techniques as holding the breath and maneuvers that elevate the intra-abdominal pressure, is also advised. Lesions that appear fixed are more likely to represent pathological conditions, whereas those that vary with provocative maneuvers are less likely to be hemodynamically relevant. Furthermore, IVUS is crucial for the accurate luminal measurements, for treatment planning, particularly to determine of an optimal landing zone for stent deploy, and for the post-deployment stent assessments [29]. IVUS has become a common tool for non-thrombotic iliac vein lesion interventions, as it demonstrates a higher sensitivity in detecting venous pathologies, particularly compression, if compared to two-dimensional or multiplanar venography. The limitations of catheter venography are especially noticeable in the anterior-posterior view, where this single-plane imaging struggles to identify lesions [31]. Furthermore, increasing use of the IVUS reduces the radiation exposure by limiting the radiation time.

## 3. Treatment of Patients with Extra-Pelvic VVs of Pelvic Origin

The treatment of extra-pelvic VVs of pelvic origin remains a topic of debate in the literature. There are two approaches to how to treat these VVs. The first one, often referred to as the “top-down” strategy, involves addressing first the pathology of the pelvic veins. This can include pelvic vein embolization or treating the iliac or renal vein compression or post-thrombotic obstruction, particularly if they are identified as the underlying cause of PeVD. This method aims at correcting the hemodynamic abnormalities that contribute to the development of pelvic symptoms and extra-pelvic VVs of pelvic origin [20].

The second approach, known as the “bottom-up” treatment, focuses on managing the peripheral, extra-pelvic manifestations of PeVD. The goal of this strategy is to alleviate extra-pelvic VVs through closing the incompetent pelvic escape points, perineal and vulvar varicosities, and also other lower limb VVs of pelvic origin, without direct management of the pathological pelvic veins. The rationale behind the latter strategy is to resolve the problem of extra-pelvic VVs in the genital region and lower limbs, thereby avoiding more invasive interventions on the pelvic veins that do not cause pelvic symptoms [20]. This approach aligns with the guidelines of the ESVS on managing chronic venous disease, stating that the treatment of patients with extra-pelvic VVs of pelvic origin should be tailored, based on the presence or absence of significant pelvic symptoms related to PVI [2].

### 3.1. Treatment of Patients Without Pelvic Symptoms

As previously mentioned, pelvic symptoms related to PVI have been reported in less than 10% of patients presenting with extra-pelvic VVs of pelvic origin [10,11]. Therefore, the 2022 ESVS guidelines suggests local procedures for VVs and related pelvic escape points in patients with VVs of pelvic origin without pelvic symptoms as the initial therapeutic approach [2]. This recommendation emphasizes the importance of addressing local superficial pathologies first, while avoiding more invasive intra-pelvic interventions, unless such procedures are clinically justified.

Since most patients with extra-pelvic VVs of pelvic origin are asymptomatic in terms of pelvic symptoms, minimally invasive procedures using the bottom-up approach are usually sufficient, while pelvic vein treatment in most of the cases is unnecessary. This approach offers several advantages: it is simple, avoids radiation exposure and the use of contrast agents, and can be done on an outpatient basis. If needed, the procedures can be repeated. However, in a case when either the bottom-up treatment fails and the vulvar and/or lower limb varicose veins recur quickly or the patient develops pelvic symptoms, management of the pelvic veins should be considered.

The most common “bottom-up” treatment methods for extra-pelvic VVs of pelvic origin include sclerotherapy, surgical ligation, and miniphlebectomy. Out of these treatment modalities, ultrasound-guided foam sclerotherapy (UGFS) is widely utilized for managing VVs of the lower limbs [20]. This technique enables an effective elimination of both incompetent tributaries and perforating veins, while the procedure is minimally invasive, and the treatment can be precisely tailored to the patient-specific anatomy of incompetent veins [32]. The pelvic escape points behave similarly to the perforating veins, connecting the pelvic veins and their tributaries to the superficial veins of the vulva, perineum, and lower limbs. Because of their morphological characteristics (these blood vessels are often tortuous and narrow), UGFS is particularly effective for their obliteration. Since sclerosing foam migrates from the injection site, the use of UGFS for pelvic escape points facilitates the closure of incompetent perivaginal venous plexuses. It can be beneficial for patients with coexisting dyspareunia related to such pathological veins. In order to facilitate the visualization of the pelvic escape points, UGFS is typically performed in the anti-Trendelenburg position (Figure 10). To enhance the patient’s comfort and reduce pain, an anesthetic cream (e.g., containing a combination of prilocaine and lidocaine) is applied topically approximately 45 min before the procedure. The entire treatment is performed under ultrasound guidance, which facilitates the precise and accurate injection of sclerosing foam. Also, migration of foam and filling of all incompetent veins is clearly visible in the ultrasonography. Identifying and eliminating all pelvic escape points is crucial to minimize the risk of recurrence, to obtain good and long-lasting results. A complete closure of target pelvic escape points not only improves clinical outcomes (Figure 11 and Figure 12), but also augments patients’ satisfaction [20,33]. An alternative treatment for pelvic escape points and related extra-pelvic varicose veins of pelvic origin is the surgical ligation combined with miniphlebectomy [34]. This approach is particularly suitable for wide (>1 cm) and short pelvic escape veins.

The efficacy of the bottom-up treatment for extra-pelvic VVs of pelvic origin has been documented in the literature. In the 24-month prospective follow-up study involving seven patients who underwent sclerotherapy for vulvar VVs reported no recurrences of VVs after the treatment. The procedure was generally well tolerated, with no serious adverse events. Only one patient experienced a transient redness of the vulva [35]. In another prospective study, 59 patients with VVs of pelvic origin in the inguinal region (30.5%), vulva (20.3%), perineum (25.5%), and buttocks (23.7%) were treated with sclerotherapy. The results were excellent in 32.6% of patients, good in 46.1%, and satisfactory in 19.1%, while unsatisfactory results were observed in only 2.2% of the patients [36]. A prospective study involving 273 patients with extra-pelvic VVs of pelvic origin assessed the outcomes of pelvic escape point surgical ligation as the treatment method. At the 1-year follow-up, the recurrence of VVs was reported in only 2.2% of the patients. The intervention was associated with minimal complications, as no cases of deep vein thrombosis, pulmonary embolism, hemorrhage, superficial vein thrombosis, wound infection, or neuralgia were reported. However, one patient experienced bleeding in the groin, which required immediate surgical intervention and prompted a modification of the surgical technique [34]. Another prospective study, involving 44 patients diagnosed with vulvar VVs of pelvic origin, evaluated outcomes of two bottom-up treatment approaches: sclerotherapy (n-12) and miniphlebectomy (*n* = 12). At the 1-year follow-up, sclerotherapy demonstrated consistent therapeutic and cosmetic benefits in 83% of patients. However, two patients experienced the recurrence of VVs 2 and 3 months after the procedure, which was attributed to subsequent pregnancies. The follow-up period for patients who underwent miniphlebectomies was extended to 3–8 years, during which no recurrences of vulvar VVs were observed. Both sclerotherapy and miniphlebectomy for vulvar VVs were free of complications. Furthermore, no patients who underwent local treatment for VVs of pelvic origin developed or experienced worsening of PVI-related pelvic symptoms [13]. Another retrospective study, involving 785 patients, reported an improvement in 78.7% of patients after sclerotherapy, with the mean follow-up duration of 4.1 ± 1.4 years [22].

The routine treatment of PVI in patients with extra-pelvic VVs of pelvic origin has not been well established in the literature. The efficacy of the “top-down” approach for the management of such VVs remains to be proven [37,38,39,40]. In the recent studies, disappearance of the extra-pelvic VVs of pelvic origin after coil embolization and top-down sclerotherapy was achieved in only 12–20% of patients [39,40]. Another study reported mild to moderate improvement of extra-pelvic VVs of pelvic origin in 51% of patients after pelvic vein embolization, with no cases of significant improvement [38]. Hence, approximately 80% of patients required additional “bottom-up” treatment to address vulvar or lower limb VVs [38,40]. Another prospective study, examining 47 patients who underwent pelvic vein embolization, found that only 12% did not require further treatment for their extra-pelvic VVs. Successful outcomes were observed mostly in patients with vulvar VVs, with 88% in this subgroup experiencing remission of varicosities. These findings emphasize that while the embolization of pelvic veins can be effective, its efficacy can be limited to specific subgroups of patients [39]. Figure 13 illustrates the venography of the pelvic escape points in a patient following coil embolization and “top-down” sclerotherapy for the treating of incompetent pelvic veins.

It has also not yet been proven whether performing the embolization of incompetent pelvic veins prevents a recurrence of lower limb VVs. Of note, the prospective study evaluating 24 patients who underwent pelvic vein embolization with simultaneous miniphlebectomy of the extra-pelvic VVs of pelvic origin reported recurrence of lower limb varicosities in 54.4% of the patients at the 3-year follow-up [37]. By contrast, another study evaluating pelvic vein embolization accompanied by miniphlebectomy of lower limb VVs recorded recurrence of lower limb VVs in only 4.16% of the patients at the 4-year follow-up, while lower limb reticular veins and telangiectasias were observed in 58.33% of the cases [41].

### 3.2. Treatment of Patients with Pelvic Symptoms

Treatment options for PVI-related pelvic symptoms primarily depend on the underlying pathophysiology, as well as on the patient’s preference.

The most common cause of CPP of venous etiology is primary pelvic venous incompetence, particularly involving the ovarian veins. In such cases, pelvic vein embolization is the treatment of choice [2,3]. A percutaneous embolization of the pelvic veins leads to the shrinking of varicose veins through stopping retrograde blood flow in the compromised blood vessels (Figure 14). In patients with PeVD with coexisting pelvic symptoms this procedure demonstrates high technical and clinical success rates, with relatively infrequent adverse events. The systematic review of 14 prospective studies reported that after the intervention, 68.3–100% of patients experienced some degree of symptomatic improvement, while 0–31.7% reported no change in their symptoms, and 0–4.1% experienced symptom worsening. Among those initially reporting symptom improvement, 0–18.2% experienced recurrence of the symptoms within 4 to 12 months [42]. In another systematic review, comprising 21 prospective case studies and one randomized controlled trial, early and substantial pain relief was observed in 75% of women. Despite a high rate of technical success, the failure rate ranged from 6% to 32%. Among patients who initially reported symptom improvement, up to 18.2% experienced symptom recurrence within 4 to 12 months [43].

Currently, there is no standardized treatment protocol for embolization of pelvic veins. Various embolic materials are employed, including metallic coils, vascular plugs, sclerosing agents, gelatin sponges, and cyanoacrylate adhesive glue. To achieve complete occlusion, these devices and pharmaceutical agents can be used alone, or in combination. Comparative studies evaluating different embolization agents are lacking, and their selection is often guided by physician preference [44]. Guidelines from leading vascular societies recommend a combination of transcatheter sclerotherapy with metallic coils as the treatment option for pelvic congestion syndrome. Still, this recommendation is based on a moderate level of evidence. The review conducted by Brown et al. reported similar improvements of symptoms in patients with pelvic congestion syndrome treated with coils, sclerosing agents, or combined embolization methods, although outcomes were slightly less favorable for those undergoing glue-based embolization [42,45,46].

In cases of a proven hemodynamically relevant iliac vein stenosis, stenting remains the standard treatment modality (Figure 15). It is important to note that only a limited number of patients suffering from non-thrombotic iliac vein lesions will clinically benefit from stent implantation, which highlights the importance of careful patient selection within this group. This is due to the fact that anatomical compressions can be found in as many as 70% of asymptomatic individuals, suggesting that a narrowing of this vein can be an incidental finding. Nonetheless, in many patients with PeVD, the symptoms can be caused by non-thrombotic iliac vein stenosis and neglecting this particular patient group can lead to inadequate or unsuccessful clinical outcomes [47].

For the treatment of these lesions, dedicated venous stents are basically recommended over other stents, due to their enhanced flexibility and a higher radial force [48]. Moreover, the proper sizing (i.e., both diameter and length) of the stent is of great importance to prevent stent migration. A proper stent should be long enough to reach a distal landing zone in the external iliac vein to prevent stent migration. Moreover, the proximal landing zone should be carefully determined, using both IVUS and venography, to ensure adequate stenting of the compression point, with minimal jailing of the contralateral CIV [49].

In cases of a combined ovarian vein reflux and iliac vein outflow obstruction, the optimal treatment approach remains a topic of debate. Some experts advocate for addressing the pelvic venous outflow lesions first, reserving treatment of the ovarian vein reflux for cases in which the symptoms persist [50]. Notably, a complete symptom resolution was reported in 76% of women with PVI caused by iliac vein stenosis following iliac vein stenting alone [51]. However, another study found the symptom resolution in only 16.6% of patients following stenting of the left CIV without ovarian vein embolization. By contrast, a combination of the iliac vein stenting and ovarian vein embolization resulted in symptom relief in 83.4% of patients [52]. Further comparative studies are needed to determine the most effective treatment approach.

Finally, it should be emphasized that most patients presenting with PVI-related pelvic symptoms and extra-pelvic VVs of pelvic origin would need additional treatment for extra-pelvic VVs after pelvic vein interventions [39,40].

## 4. Conclusions

Management for extra-pelvic VVs of pelvic origin should be individualized, mainly based on the patient’s symptoms. Bottom-up techniques could be considered as an initial therapeutic approach, to avoid unnecessary and more invasive treatments such as pelvic vein embolization or iliac vein stenting.

## Figures and Tables

**Figure 1 jcm-14-02707-f001:**
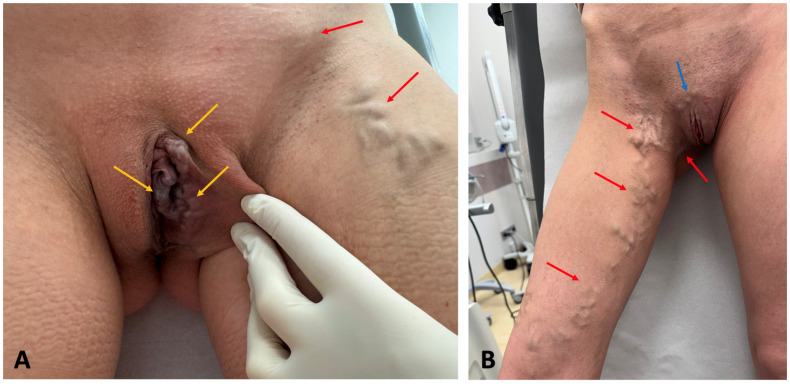
Extra-pelvic varicose veins (VVs) of pelvic origin. (**A**) Vulvar VVs (yellow arrow) and atypical VVs of pelvic origin at the inguinal area and lower extremity (red arrow); (**B**) atypical VVs of pelvic origin at the lower extremity (red arrow) and at the mons pubis (blue arrow).

**Figure 2 jcm-14-02707-f002:**
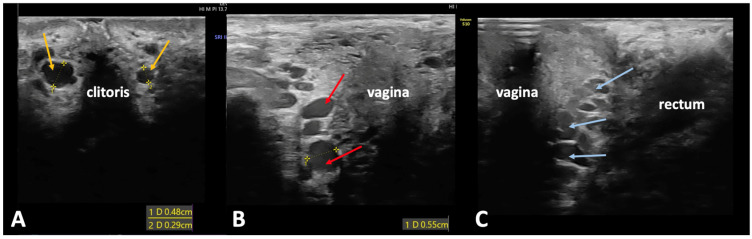
Duplex ultrasonography of the pudendal escape points. (**A**) Clitoral escape point (yellow arrows) with the deep dorsal veins of the clitoris located on both sides of the clitoris−measuring 0.48 cm in diameter on the right side and 0.29 cm on the left side; (**B**) intermediate labial escape point (red arrow) with the vein of the bulb of vestibule running along the anterior and lateral wall of the vagina−measuring 0.55 cm in diameter; (**C**) perineal escape point (blue arrow) with the perineal vein, which runs along the posterior vaginal wall within the space between the vagina and the rectum.

**Figure 3 jcm-14-02707-f003:**
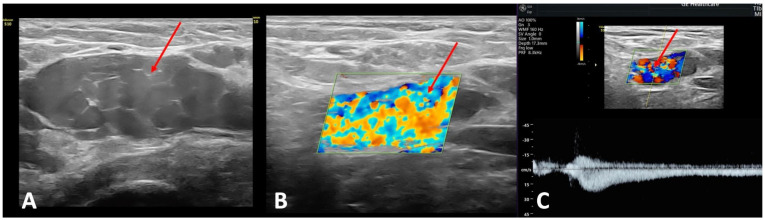
Duplex ultrasonography of the inguinal escape point (red arrows). In B−mode imaging (**A**), the escape point appears as an area of hypoechoic, spongy−like tissue. Reflux is clearly visualized in color Doppler (**B**), appearing as bidirectional or turbulent flow within the region of interest, characterized by alternating color signals (typically blue, red and yellow), indicating reversal or disturbance of venous flow. This hemodynamic abnormality is further confirmed with pulsed−wave Doppler (**C**), where a retrograde flow component lasting longer than 7 s can be observed, supporting the diagnosis of venous reflux at the inguinal escape point.

**Figure 4 jcm-14-02707-f004:**
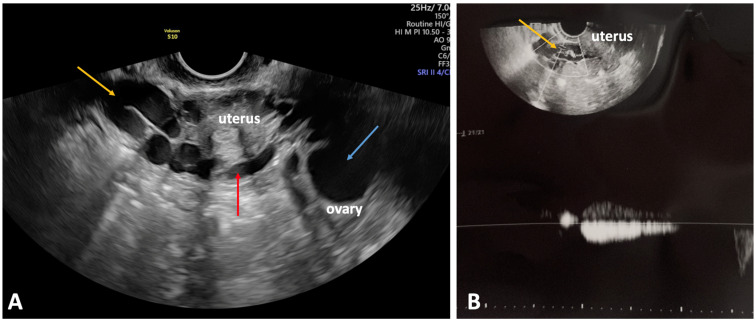
Transvaginal ultrasonography of a patient with pelvic venous disorders and pelvic symptoms. (**A**) Gray-scale ultrasound shows varicose veins of the periuterine venous plexus (yellow arrow) adjacent to the uterus, connected to a vein traversing the uterine body (red arrow). A unilocular ovarian cyst is also identified in the right adnexa (blue arrow). (**B**) Doppler ultrasonography demonstrates reflux (duration: 2 s) in the periuterine venous plexus (yellow arrow) following distal manual compression in the ipsilateral iliac fossa, indicating venous insufficiency.

**Figure 5 jcm-14-02707-f005:**
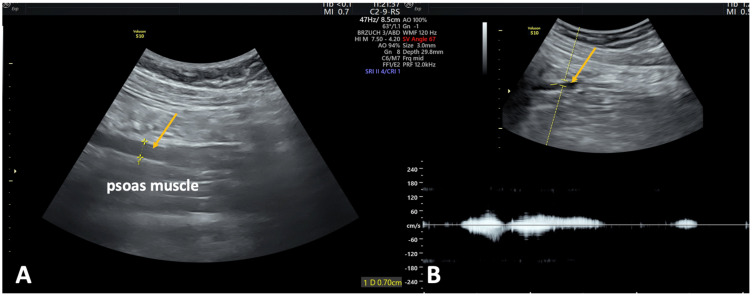
Transabdominal ultrasonography in a patient with pelvic venous disorder and pelvic symptoms. (**A**) Gray−scale image reveals a dilated left ovarian vein (yellow arrow) measuring 7.0 mm in diameter, coursing anteriorly to the psoas muscle− an imaging feature suggestive of ovarian vein insufficiency. (**B**) Combined color and spectral Doppler ultrasonography demonstrates venous reflux in the same left ovarian vein (yellow arrow), lasting approximately 2 s, triggered by manual compression in the ipsilateral iliac fossa. These findings confirm the presence of hemodynamically significant retrograde flow.

**Figure 6 jcm-14-02707-f006:**
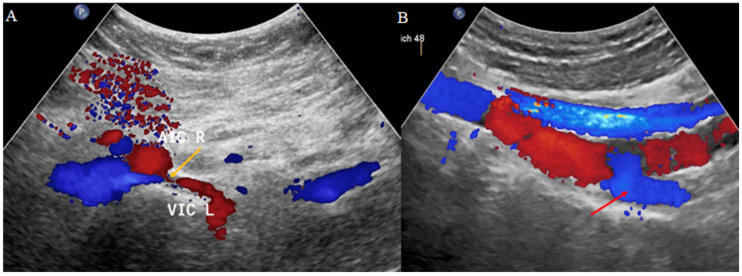
Transabdominal duplex ultrasonography in a patient with May–Thurner syndrome. (**A**) Color Doppler ultrasound shows compression of the left common iliac vein (VIC L) by the overlying right common iliac artery (AIC R) (yellow arrow), consistent with May–Thurner anatomy. The venous segment appears narrowed at the point of arterial crossing. (**B**) Color Doppler imaging demonstrates reflux (red arrow) in the left internal iliac vein, suggesting venous hypertension and collateral flow secondary to proximal outflow obstruction.

**Figure 7 jcm-14-02707-f007:**
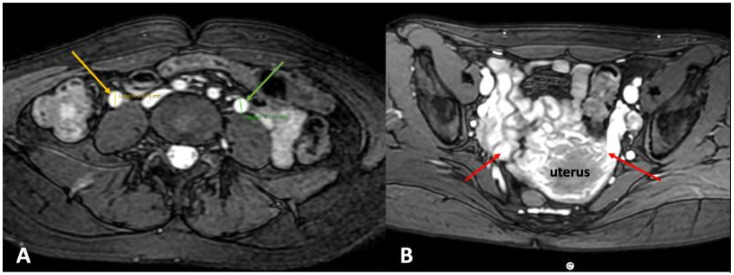
Magnetic resonance venography in a patient with pelvic venous disorder. (**A**) Axial view showing dilated ovarian veins: the left ovarian vein measures 9.1 mm (green arrow) and the right ovarian vein measures 9.9 mm (yellow arrow). (**B**) Coronal view reveals prominent varicose veins within the periuterine venous plexus (red arrows), consistent with pelvic venous congestion.

**Figure 8 jcm-14-02707-f008:**
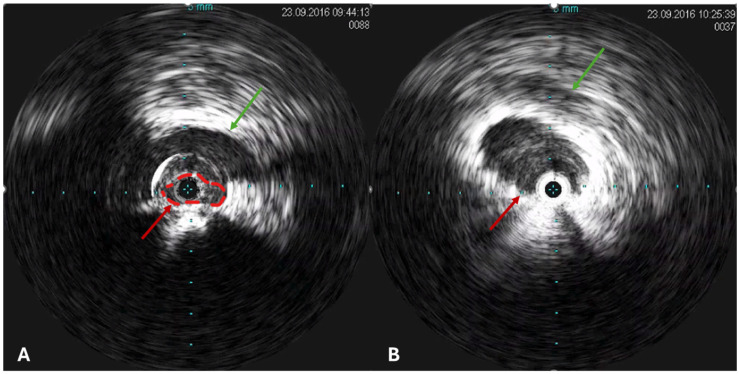
Intravascular ultrasound (IVUS) in a patient with May–Thurner syndrome. (**A**) IVUS image prior to stenting demonstrates significant compression of the left common iliac vein (red arrow) by the overlying right common iliac artery (green arrow), with eccentric narrowing of the vein lumen. (**B**) IVUS image at the same location following venous stent placement shows restored luminal patency and resolution of the extrinsic arterial compression (red arrow: expanded stent; green arrow: right common iliac artery).

**Figure 9 jcm-14-02707-f009:**
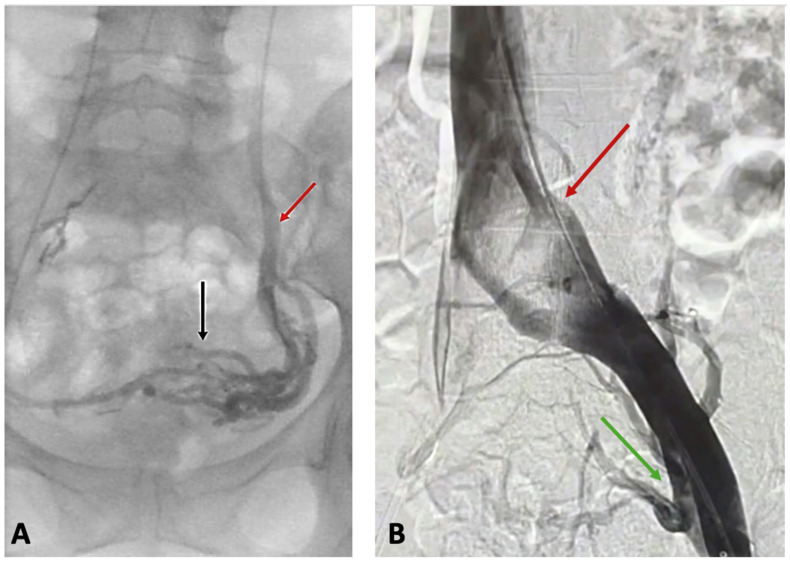
Catheter venography findings in patients with pelvic venous disorders. (**A**) Selective catheter venogram of the left ovarian vein reveals significant dilation and incompetence (red arrow), with contrast reflux into the periuterine venous plexus (black arrow) and prominent pelvic varicosities. (**B**) Catheter venogram in a patient with May–Thurner syndrome demonstrates focal narrowing at the compression point of the left common iliac vein (red arrow), with reflux into the internal iliac vein (green arrow) and evidence of collateral venous pathways.

**Figure 10 jcm-14-02707-f010:**
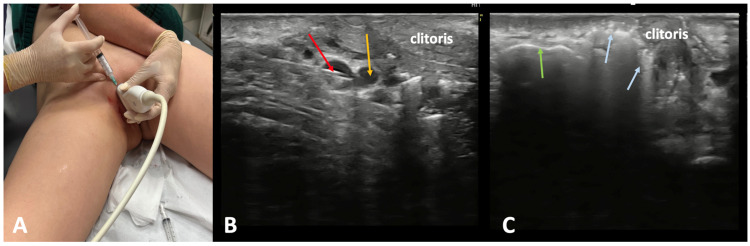
Ultrasound-guided foam sclerotherapy of the clitoral pelvic escape point. (**A**) Access to the clitoral pelvic escape point. (**B**) Puncture of the clitoral pelvic escape point (yellow arrow) visualized in ultrasonography; the needle is marked with the red arrow. (**C**) The clitoral pelvic escape point (blue arrow) and the external pudendal vein (green arrows) filled with foam.

**Figure 11 jcm-14-02707-f011:**
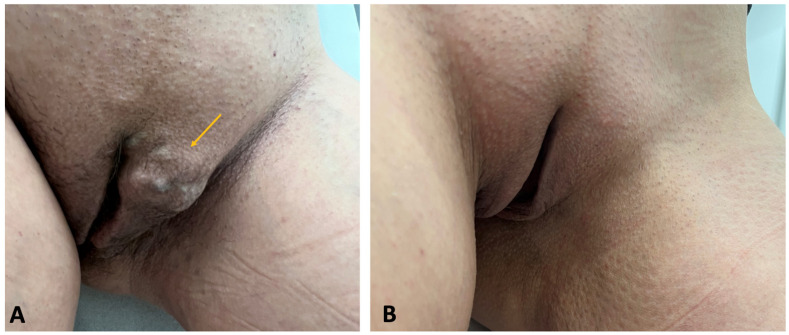
Patient with vulvar varicose veins (VVs) of pelvic origin. (**A**) Patient presenting with vulvar VVs (yellow arrow) associated with an incompetent clitoral pelvic escape point before the treatment. (**B**) The same patient 12 months after ultrasound-guided foam sclerotherapy for the clitoral pelvic escape point and vulvar VVs.

**Figure 12 jcm-14-02707-f012:**
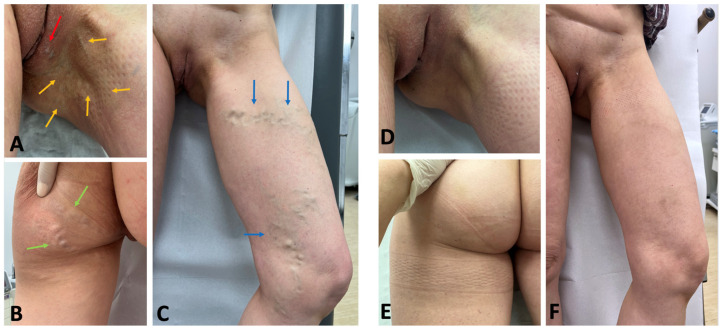
Patient with extra-pelvic varicose veins (VVs) of pelvic origin before the treatment (**A**–**C**) and 12 months after ultrasound-guided foam sclerotherapy of the pelvic escape points and related VVs (**D**–**F**). This patient presented with vulvar VVs (red arrow), VVs of pelvic origin at the buttock (green arrows), and lower limb varicose veins of pelvic origin located on the medial (yellow arrows) and anterior (blue arrows) aspects of the thigh.

**Figure 13 jcm-14-02707-f013:**
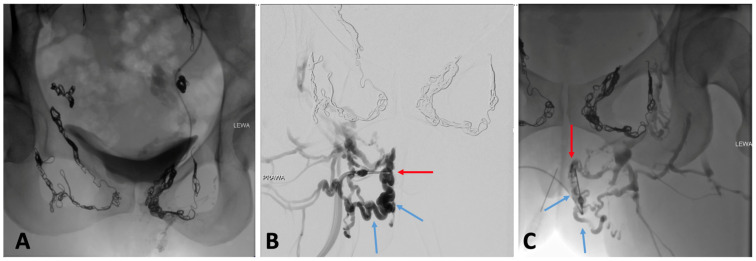
Venographic evaluation of a patient with pelvic symptoms and extra-pelvic varicose veins (VVs) of pelvic origin following pelvic vein embolization. (**A**) Post-embolization venogram after coil embolization and sclerotherapy of the bilateral ovarian, uterine, obturator, and pudendal veins, showing treated vessels with retained contrast and coils. Despite this, the patient reported persistent extra- pelvic VVs of pelvic origin. (**B**) Direct puncture venography of the right perineal escape point (red arrow) reveals reflux into VVs of the vulva and perineum (blue arrows), indicating incomplete „top-down” treatment of distal drainage pathways. (**C**) Similar findings on the left side: persistent reflux through the left perineal escape point (red arrow) with opacification of residual vulvar Vvs (blue arrows), confirming failure of the initial “top-down” embolization approach to completely occlude the outflow routes.

**Figure 14 jcm-14-02707-f014:**
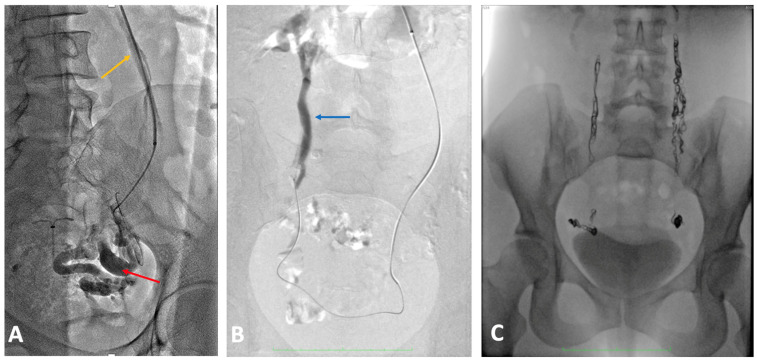
Venography of a patient presenting with pelvic symptoms of venous origin. (**A**) Dilatated and incompetent ovarian vein (yellow arrow) causing varicose veins of periuterine venous plexus (red arrow). (**B**) Dilatated and incompetent right ovarian vein (blue arrow) filled with contrast using the cross-over technique. (**C**) Venography after embolization of both ovarian veins.

**Figure 15 jcm-14-02707-f015:**
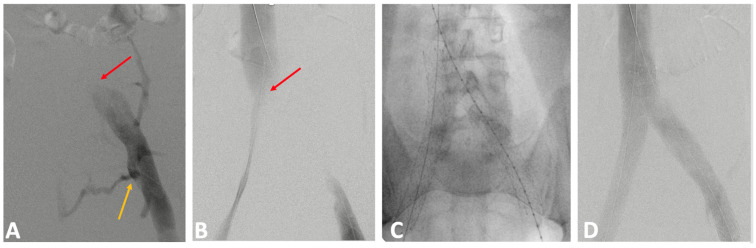
Patient with bilateral common iliac vein (CIV) compression. (**A**) Preprocedural venogram reveals a hemodynamically relevant left CIV compression (red arrow), with reflux in the internal iliac vein (IIV) (yellow arrow). (**B**) Compression of the right CIV (red arrow). (**C**) Stenting performed using the double-barrel technique, followed by intravascular ultrasound check after stent deployment. (**D**) Final venogram demonstrates the disappearance of the collateral veins and reflux in the IIV.

## Data Availability

No new data were created or analyzed in this study. Data sharing is not applicable to this article.

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
