# Peer review of "Management of Extra-Pelvic Varicose Veins of Pelvic Origin in Female Patients"

_jcm, 2025, doi:10.3390/jcm14082707_

Round 1
Reviewer 1 Report
Comments and Suggestions for Authors
Everithing is OK
The management of extra-pelvic varicose veins of pelvic origin in female patients is an important and relevant topic due to the high prevalence and significant symptomatic impact of PeVD in women. Pelvic varicose pathology has been extensively studied, and the review presented by the authors is well-designed and appropriately focused on extra-pelvic varices. The paper is well-developed, practical, and of great interest to readers. The selection of the topic is highly relevant, and the discussion provides valuable insights for both clinicians and researchers in the field.
Author Response
Dear Reviewer,
Thank you very much for your kind and thoughtful review. We truly appreciate your feedback and support!
Reviewer 2 Report
Comments and Suggestions for Authors
The present review is comprehensive, well written, starting with a brief but highly relevant background in anatomy and physiology, continuing with the diagnostic methods that are detailed and significant for immediate clinical practice, including contemporary controversies. The manuscript presents the whole range of treatment methods, based on actual evidence, supported by convincing clinical cases.
Author Response
Dear Reviewer,
Thank you very much for your nice comments and the review. We appreciate your feedback.
Reviewer 3 Report
Comments and Suggestions for Authors
Dear Authors,
You raised an important topic of management of extra-pelvic varicose veins of pelvic origin. There is a lack of good-quality studies in this field, there is no standardization of therapeutic methods. The whole "Pelvic Congestion Syndrom" or "Pelvic Vein Disorders" is still a "white chart of phlebology in the matter of EBM. What is even more, there is a serious issue with the optimal diagnostic pathway or assessment of symptoms and its impact on life quality (at the moment there is only one available tool dedicated for PCS, but its validation showed that it is not well-designed - doi: 10.24425/fmc.2023.145914).
It is highly beneficial that this manuscript covers multiple areas of issues related to therapy strategies, but also emphasizes the key role of good-quality examination and diagnostics of the patients, before they qualify for invasive treatment.
I only must disagree with the last conclusion - your manuscript does not allow to recommend the use of bottom-up technique as initial. You should rephrase, suggesting that this technique should be considered, but without suggesting its priority, as no big-data studies could support this opinion.
I do not have any further requests or questions.
Kind regards
Author Response
Dear Reviewer,
Thank you very much for your thoughtful and constructive review. We truly appreciate your recognition of the importance of this topic and the current gaps in evidence-based management of extra-pelvic varicose veins of pelvic origin. As you rightly pointed out, the lack of high-quality studies, standardization in therapeutic approaches, and validated diagnostic tools underscores the need for further research and consensus in this field.
We also value your positive remarks on the manuscript’s comprehensive approach to addressing both therapeutic strategies and the importance of thorough diagnostics prior to invasive treatment.
Regarding your comment on the concluding statement about the bottom-up technique, we will revise the conclusion accordingly to clarify that the bottom-up approach could be considered as an initial method.
Thank you once again for your valuable input and support.
Reviewer 4 Report
Comments and Suggestions for Authors
Management of extra-pelvic varicose veins of pelvic origin in female patients
General comments
The paper is very well written and interesting, covering all the aspects of PVI.
I would add some words about the clinical presentation of these VV patients: usually, they present moderately enlarged varices (C2), not causing skin damage but accompanied by diffuse limb pain/heaviness, typically related to hormonal phases; you can guess the diagnosis by telephone talk, due to its significance. A trial with stockings may be very effective in treating symptoms. There is a high probability of hormones passing into the leg circulation through the leaking points, as suggested by Asciutto, G. et al. (Oestradiol Levels in Varicose Vein Blood of Patients with and without Pelvic Vein Incompetence (PVI): Diagnostic Implications European Journal of Vascular and Endovascular Surgery, Volume 40, Issue 1, 117 – 121).
From the diagnostic point of view, the importance of the Valsalva maneuver, although cited, could be strongly stressed (and even explained) as the most effective way of revealing refluxes together with manual abdominal compression. A standing position in VV analysis would be preferable.
Specific Comments
Please control words spacing in line 41, 44, 48.
The sentence in lines 39-41: “These varicosities, which are one of clinical manifestations of pelvic venous disorders (PeVD), resulting from pelvic vein incompetence (PVI) and pelvic venous hypertension”, needs a verb.
Author Response
Dear Reviewer,
Thank you very much for your kind and encouraging feedback. We are pleased to hear that you found the paper well written and interesting, and we greatly appreciate your insightful suggestions for improvement.
We fully agree that the clinical presentation of patients with pelvic vein incompetence (PVI)-related varicose veins deserves additional emphasis. Your suggestion to describe the clinical pattern- including moderately enlarged VVs (C2, based on CEAP classification), not causing skin damage but accompanied by diffuse limb pain/heaviness, typically related to hormonal phases- is very valuable. We will integrate this into the manuscript. We also appreciate your reference to the work by Asciutto et al. regarding the hormonal influence, and we will cite this appropriately in the revised version.
Regarding diagnostics, we agree that the Valsalva maneuver, in conjunction with distal manual compression in the ipsilateral iliac fossa, is crucial in revealing reflux in the pelvic veins, and we included it in the subchapter dedicated transvaginal ultrasonography, as well as in transabdominal ultrasonography and ultrasonography of pelvic escape points. We also emphasized the importance of performing vein assessments in the anti- Trendelenburg position, as it offers the most accurate hemodynamic evaluation.
Thank you also for the specific corrections noted. We will address the spacing issues on lines 41, 44, and 48 and revise the sentence on lines 39–41 to include the missing verb for clarity.
Your comments have been extremely helpful in refining and improving the manuscript, and we are grateful for your thoughtful review.
Reviewer 5 Report
Comments and Suggestions for Authors
A comprehensive and thoughtful review of extra-pelvic varicose veins. Provides background and insight about current treatment paradigms. Would benefit from a few more references in some areas.
Comments on the Quality of English LanguageWould be improved with minor editing for readability.
Author Response
Dear Reviewer,
Thank you very much for your positive and encouraging feedback. We appreciate your recognition of the manuscript as a comprehensive and thoughtful review of extra-pelvic varicose veins, and your acknowledgment of the background and insights provided into current treatment paradigms.
We added a reference regarding a high probability of hormones passing into the leg circulation through the leaking points, by Asciutto, G. et al, and we have now 52 references in total.
Regarding the language and readability, we have carefully edited the manuscript to improve clarity and flow, addressing minor issues to enhance overall readability.
Thank you once again for your helpful comments and your time in reviewing our work.